# Between the Lines: An Exploration of Online Academic Help-Seeking and Outsourced Support in Higher Education: Who Seeks Help and Why?

Lorraine Delaney [1,*], Mark Brown [2] and Eamon Costello [3]

1   School of Human Development, Institute of Education, Dublin City University, Drumcondra,
    D09 V209 Dublin, Ireland
2   School of Policy and Practice, Institute of Education, Dublin City University, Drumcondra, D09 V209 Dublin,
    Ireland; mark.brown@dcu.ie
3   School of STEM Education, Innovation and Global Studies, Institute of Education, Dublin City University,
    Drumcondra, D09 V209 Dublin, Ireland; eamon.costello@dcu.ie
*   Correspondence: lorraine.delaney@dcu.ie

**Abstract:** The growth in online higher education has seen the 'unbundling' of some services as universities have partnered with private companies in an effort to enhance their services. This paper explores university students' use and perceptions of the third-party online learning support platform, Studiosity, at Dublin City University. Studiosity was engaged to support undergraduate and postgraduate distance students, by offering support beyond existing campus-based services. This research employs a primarily inductive research design drawing on data collected through the third-party provider (2018–2020), supplemented by an in-house online survey (2019). Students were overwhelmingly positive about Studiosity. Postgraduate students, arguably students with good academic skills, used the service more than first-year undergraduate students. However, first-year undergraduates, a group the literature suggests are reluctant users of institutional support, were also strong users. Questions emanating from postgraduate students demonstrated expedient help-seeking. First-year undergraduate students were more concerned with explanations to help their understanding in order to persist with their studies. This paper posits that all other things being equal, those who already have strong academic capital will be the greatest users of academic support services. Proactive, less formal academic support strategies to encourage use by those who need help most, remain critical.

**Keywords:** student; academic; support; Studiosity; online

## 1. Introduction

The growth in online higher education has seen the 'unbundling' of some services as universities have partnered with private companies in an effort to enhance student support [1]. Unbundling is defined by Swinnerton et al. [2] (p. 218) as the 'disaggregation of educational provision into its component parts'. This process can involve multiple stakeholders using digital approaches. The increasing influence of commercial providers in the online learning landscape sets the wider context in which this paper examines third-party outsourced academic support services.

Within universities, the provision of enhanced academic support and development for students has become ubiquitous. The main reason given for this focus is to enhance student retention and success [3–6]. Linking student success and institutional quality is becoming increasingly prevalent, and the role of academic support and development services is now a taken-for-granted part of effective student support. Yet, relatively little is known about the nature of academic help-seeking in general [3], and most especially in an Irish context.

This paper explores university students' use and perceptions of the third-party online learning support platform, Studiosity, at Dublin City University (DCU), a dual-mode

university with approximately 18,000 full-time students, largely made up of campus-based undergraduate and postgraduate students, and around 1800 part-time students studying online. Developed in Australia, Studiosity claims to support 1.6 million learners in over 150 institutions worldwide.

In 2018, a decision was taken to pilot Studiosity for DCU, managed centrally by the Open Education Unit (OEU), a department with a long history of widening access to university education through distance learning. This decision was primarily made because distance students found it difficult to access existing academic support and development services primarily designed for campus-based learners. While online meetings were possible, it was difficult to obtain appointments outside of normal working hours that suited the study habits of part-time online learners. Student support and development services were generally provided during the day, with few evening and no weekend appointments.

This study takes place at a time of increased demand for academic support in higher education coupled with the increased presence of third-party providers in that space. Despite this, there is a dearth of analysis on academic help-seeking at university level [7,8]. Much of the literature related to the use of Studiosity consists of brief case studies featured on the provider's website with limited peer-reviewed research. Employing an inductive research design, this paper explores the experience of two specific groups of students using Studiosity: first-year undergraduate (*n* = 380) and post-graduate (*n* = 401) students. What unique self-expressed learning support and development needs do these groups have? What can we learn from the questions they ask and the types of academic help they seek? The aim of this research is to inform academic support service provision in the future.

This article has five remaining sections. The next section provides a review of relevant literature. The third section outlines the materials and methods, including the research questions. The fourth section presents an analysis and discussion of the findings while the final section offers some concluding remarks and identifies areas for further research.

## 2. Literature Review

This review examines the literature regarding three aspects of academic support. Firstly, we explore the role of academic support, why institutions provide it and to whom. Secondly, we examine how academic support is provided. Finally, we explore the nature of academic help-seeking, identifying the two dominant forms: instrumental and expedient.

### 2.1. The Role of Academic Support

The literature on the role of academic support primarily focuses on its ability to enhance retention and persistence. Some studies focus on enhanced academic support for undergraduate students [4,7,9–14]. Often this support is targeted at underrepresented first-year students who are poorly prepared for university and may enter the institution with limited background academic capital to the extent that they may have little specific knowledge about how university 'works' [15]. The role here is one of attempting to level the playing field to address structural inequalities and support successful completion.

Other studies focus on supporting the persistence of all students equally, undergraduate and postgraduate [5,8,16–18]. These studies reason that undergraduate and postgraduate students face similar challenges when transitioning to university [18–20]. But postgraduate students have, by definition, successfully made that transition. Additionally, there is less evidence that high attrition rates are problematic in postgraduate education. Such studies appear at odds with those identifying a greater need in the undergraduate, underrepresented cohort.

### 2.2. Provision of Academic Support

Academic support can be either formal, accessed within or through the institution, or informal, accessed outside the institution through social contacts. The effectiveness of academic support is variably related to its responsiveness [5], timing [16], flexibility [21], quality [22], and perhaps most importantly, uptake [12,23].

Within the institution, academic support is often offered on campus and in person. The advantages of this Socratic style are well documented [7,12,13]. Students develop transferable skills such as problem posing, problem solving, and critical thinking. However, there are problems with this type of support. It is often restricted to office hours, and so does not serve the needs of part-time/distance students [6,16]. It can also be presented as more formal as students make appointments and wait to be seen. This delay in obtaining help may be off-putting for less confident learners. Working-class first-year undergraduate students are less likely to initiate formal help-seeking [17,24], and are more likely to ask for help from peers over professors [4,17]. While peer support is hailed as valuable for under-represented undergraduate students [17], there are inherent dangers; peer support may be inaccurate or incorrect. It may therefore be best not to rely on peers for hard information.

Increasingly, technology is playing a role in academic support provision in higher education [5,6,8,16,25]. Online support can be provided by staff within the institution, or it can be outsourced to a third party, a practice known as 'unbundling' or 'distributed' support. Benzie and Harper [10] (p. 645) argue that academic support in more distributed learning environments is problematic because '…they contribute to a context for writing that is unbounded, generic, and fragmented'. This criticism is echoed by Gurney and Grossi [26], who state that the capacity of third-party providers to develop students' autonomy as academic communicators is restricted. They tend to 'fix' students' work, rather than encouraging them to think autonomously. Rambiritch [13] (p. 58) also sees such consultations as editing rather than dialogue and calls for 'evaluating the quality of such contributions to student development'.

Yet technology can and does support dialogue, both synchronous and asynchronous. Amador and Amador [9] report on the success of Facebook in generating a sense of community through online interactions. A study of *Live chat* by Broadbent and Lodge [11] highlighted how responsive the service was, particularly for online students, who felt more cared-for and connected to the institution due to the service. Dollinger et al. [16] also report positively on the *Connect Live* service offered by Studiosity, where students were connected in real time with a human tutor, to support their learning. However, while the potential exists for dialogue in the online environment, the extent to which dialogue or, in a related sense, instrumental support, takes place is not explored.

There are challenges with third-party support. There can be problems with acceptance of such provision within the institutional culture, problems with different tutors giving different advice, and overly positive feedback being provided to students by the third party [4,6,10]. Notwithstanding such criticism, the availability of online support outside regular office hours is particularly welcomed by students, especially students who are working, or are living in time zones which differ from the institution [16]. Evaluations of online support generally receive high student satisfaction ratings [8,16].

### 2.3. Academic Help-Seeking

Academic help-seeking relates to students seeking help with academic learning [22,27,28], primarily in relation to understanding course content [29]. Within the literature, help-seeking is broadly divided into two categories: instrumental and expedient [27].

While instrumental help-seeking often takes the form of dialogue [10,13], it more generally relates to the student seeking to solve the problem themselves by asking for explanations of concepts they do not understand [22].

Conversely, expedient help-seeking is seen to be asking for help with something one is/should be capable of solving oneself [29]. What one is capable of solving oneself is likely to be different for different student cohorts, though this point is not fully addressed in the literature.

While Calarco [29] identifies that both methods of help-seeking lead to improved academic performance, this is disputed by Golann and Darling-Aduana [30], who point out that with expedient help-seeking, students become over-reliant on others, limiting higher-order cognition. Expedient help-seeking is therefore regarded as less likely to

lead to deep learning, and so is sometimes regarded as of less value than instrumental help-seeking [10,13,22,26,31].

The literature highlights how first-generation undergraduate students prefer to engage in less formal help-seeking, with a preference for resources that are convenient, reliable, easy to access and online [17,32]. Certainly, to overcome any internalised stigma against help-seeking [33], and reframe the way students view it [17,34], the importance of informal style, accessible help is important. While students with high levels of academic capital may be able to avail themselves of this type of help through their social networks, this option is not often open to working-class students who may be the first in their family to attend college and may have few friends progressing to university. So although expedient help is seen as less valuable in the literature, for some students, expedient help-seeking may be a prerequisite to instrumental help-seeking. It may break down barriers to help-seeking and allow students to see it as a normal aspect of university study. This may be particularly important for underrepresented undergraduate students.

What Bourdieu [35–37] recognised as 'cultural or academic capital' and Lareau [38] as a 'sense of entitlement' results in students from more middle-class backgrounds having assertiveness and an ease in interacting with authority [30]. Those from more working-class backgrounds tend to operate with 'a sense of constraint' [39], showing more caution when interacting with authorities or avoiding interaction altogether. This practice persists to the extent that by the time students enter university, their help-seeking behaviour tends to be well established [27,30].

The literature calls for more evaluation of third-party academic support, in particular its potential to foster instrumental help-seeking. To date, no study has examined if or how academic help-seeking differs between undergraduate and postgraduate students and what we can learn about help-seeking from their pattern of engagement with Studiosity and the questions they ask. This paper seeks to address these current gaps in the literature.

## 3. Materials and Methods

This study focuses on two groups of students who used Studiosity and the type of support they sought. Group one comprised first-year (FY) undergraduate online distance students ($n$ = 380), selected because they can be challenging students to retain [9], having invested relatively little in the course compared to students who are further on in their studies. Part-time, undergraduate online distance students tend to be older than traditional, full-time undergraduate campus-based students and are more likely to be from lower socio-economic backgrounds, having often delayed their participation in higher education for reasons related to social class [39–41]

The second group ($n$ = 401) comprised postgraduate (PG) online distance students. This group was a major user of the service. Hence, this study was intended to help us understand this usage better. Postgraduate students are not new to higher education as they normally already hold a level 8 honours primary degree.

While the ethnic background of students was not identified for the purposes of this study, all non-native speakers of English are required to provide evidence they meet the university's English language requirements on entry.

The Studiosity platform provided two distinct services, outlined in Table 1:

**Table 1.** Studiosity academic support services.

|  | **Writing Feedback (WF)** | **Connect Live (CL)** |
|---|---|---|
| Service | Writing submission | Online live chat |
| Offered when | 24/7 | Six days a week (Mon–Sat) |
| Response time | 24–48 h | Approx 30 min |
| Types of submissions | Essays, reports, résumé, etc., up to 5000 words | Course/subject-related questions |
| Responder | One-to-one from suitable responder/subject specialist | One-to-one from suitable responder/subject specialist |

**Table 1.** *Cont.*

|  | Writing Feedback (WF) | Connect Live (CL) |
|---|---|---|
| Additional functions |  | Collaborative whiteboard for file sharing, maths, science-related queries |
| Feedback | Written feedback on grammar, structure, readability Academic help but not answers | Oral feedback related to the question Academic help but not answers |

This research employs a primarily inductive research design. This study draws on two datasets. The first consists of monthly reports provided to DCU [by Studiosity. The second dataset is from an online in-house survey ($n = 54$). Both instruments provide quantitative and qualitative data. Ethical approval was obtained for this study from the DCU Research Ethics Committee. All participants were adults. Notably, this study took place over two academic years: 2018–2019 (pre-pandemic) and 2019–2020 (during the pandemic). All data were gathered online and anonymised.

The research sought to answer the following questions:

- What was the pattern of student cohort engagement with Studiosity?
- Why did students engage/not engage with Studiosity (i.e., what can we learn from the questions they ask)?
- What were students' perceptions of Studiosity?
- What are the implications of student engagement patterns with Studiosity?

### 3.1. Studiosity Data

The entire cohort of [DCU's distance students in each academic year ($n = 644$ in 2019, $n = 764$ in 2020) were invited to use the Studiosity service. Studiosity provided monthly reports containing both data for each specific month and cumulative data up to that point (the cumulative reports for August in each academic year contain the overall picture). Studiosity gathered information on the number of times the service was used, the type of service students used, time spent on each engagement, login time patterns by time of day and day of week, course level, subject area, type of document submitted, and student satisfaction rating, primarily quantitative data. Studiosity also provided information on the questions asked, together with student comments regarding the responses to those questions, primarily qualitative data. Student comments were provided by Studiosity for 2019/20 only.

### 3.2. Student Survey

In December 2019, the research team administered a survey, through the online survey tool Qualtrics, to all students ($n = 764$), the purpose of which was threefold:

- to remind students about Studiosity and encourage them to use it.
- to find out why they were not using Studiosity.
- to evaluate the service independently of the service provider's own evaluation.

Participation in the survey was voluntary and anonymous. The survey was based on relevant literature and timed to encourage a high response rate. Students are well established in their courses in December (the academic year starts in September), yet do not have any significant assessment commitments, such as exams. Conducting the survey in December also allowed the team to assess students' live experience with the platform, rather than asking them to recall their experience at a later point. The survey timing was also designed to assist the institution in evaluating whether to continue the service beyond the initial two-year pilot. The response rate was low ($n = 54$, 7%), though low response rates are normal for online, anonymous surveys [42].

*3.3. Data Analysis*

Quantitative data were analysed using Excel. Content analysis was deemed the most appropriate method of analysing the qualitative data as it most clearly links qualitative and quantitative approaches to data. Content analysis allows us to identify core reasons and their prevalence [43]. Within this study, transcripts of the qualitative data were jointly analysed, to guard against bias. Constant comparison was employed to identify themes and categories, producing a coding schedule and counting occurrences. Constant comparison generates relationships or patterns of similarity–difference [44]. It allows the development of a story about the data that is coherent and recognisable from the data. Content analysis facilitates the systematic description of a phenomenon, e.g., the type of question students ask in an online academic support forum/the analysis of keywords or messages to an online academic support forum. Qualitative data bring depth to our understanding, helping reveal information that might be important to the overall picture. Qualitative analysis is generally interpretative, taking a constructivist, as opposed to a realist, position on knowing. Some interesting findings emerge.

**4. Results and Discussion**

*4.1. Studiosity Data*

4.1.1. Pattern of Student Cohort Engagement with Studiosity

The service was promoted equally to all distance student cohorts. The overall outline of the engagement with Studiosity is provided in Table 2. The first notable pattern of engagement was that postgraduate students used the service more than first-year under-graduate students, even though arguably they already have strong academic skills in place. The literature suggests that those who enter university with greater academic capital are better positioned to avail themselves of academic help. The evidence from this study suggests that when academic help is provided to all students equally, those who already hold academic capital, and have experienced higher education success (i.e., those other than first-year undergraduates) are likely to be the biggest users.

**Table 2.** Students, unique users and instances of use 2018/2019 and 2019/2020 combined.

| | Number of Students in Cohort | Users | Percentage | Uses | Average No. of Uses | Max No. of Uses by Individual |
|---|---|---|---|---|---|---|
| First-year undergraduates | 380 | 74 | 19% | 451 | 6 | 17 |
| All other undergraduates | 627 | 194 | 31% | 566 | 3 | 17 |
| Postgraduate | 401 | 115 | 29% | 651 | 6 | 14 |
| Total | 1408 | 383 | | 1668 | | |

However, first-year students are strong users of the service (Table 2). First-generation students (e.g., distance undergraduate students), tend to be reluctant users of institutional help [17]. However, the evidence from this study suggests that first-year undergraduate distance students will, under favourable circumstances, avail themselves of institutionally provided academic support.

The second pattern of note related to instances of use, which were not reflective of the real number of students seeking assistance. For example, in relation to postgraduate students, there were 651 instances of use by 401 students (Table 2). Some students used the service a lot, and others not at all. The maximum number of times a postgraduate student used the service is 14 times, while the maximum number for a first-year undergraduate student was 17 times.

Regular use of a service by a small group of students can have negative consequences, particularly in a resource-constrained environment as there is often a limit to the number

of appointments available. If I receive more support, you receive less. If I am better able to ask for help, I obtain a bigger piece of the support pie. Academic help-seeking becomes a zero-sum game.

One possible solution to this situation, if it is problematic, may be to restrict the number of times a student can use the service in a particular month or over the academic year. This may also encourage students to be more careful about the quality of their questions and requests for help, with less reliance on the service over time as they develop their own academic capital and learning strategies.

Although Studiosity imposed a 5k word count for uploads, students circumvented this by uploading sections of a document separately:

> *I am preparing a document longer than the 5k limit. I had put through the first 5k words a few days before. Very little was picked up. I put through the last 5k words this time, (February 2020 PG WF)*

> *I sent one document broken into 5 lots to Studiosity yesterday. (May 2020 PG WF)*

Another strategy employed by students was to upload the same document, with the same questions, several times:

> *I put the document through again and got much more insightful comments from another Reviewer. I am glad I put the work through a second time. this might be harder to do in the university based system? (February 2020 PG WF)*

> *I was interested to see if there was a difference in opinion. (February 2020 PG WF)*

Students used the Writing Feedback service far more regularly than the Connect Live service, with 1615 submissions against 69 instances of use over the two years, echoing the findings of Dollinger et al. [16]. This may indicate a preference for non-direct engagement. This point is examined in more detail when analysing our survey results.

Instances of academic help-seeking through Studiosity increased dramatically in 2019/2020 (Table 3) from 509 instances to 1159, when the university was in lockdown due to the global pandemic. This increase cannot be explained by an increase in student numbers (+18%) nor the mode of delivery, which continued to be online. The increase may relate to the fact that most students were now working from home and perhaps had easier access to the internet and less social diversions. Students may have used the service more as they sought more connection due to the isolation of the lockdown. The increased use may also relate to greater familiarity with the platform and/or greater awareness of it. We can only speculate on the reasons for this growth in usage.

**Table 3.** Change in numbers using Studiosity from 2018/2019 to 2019/2020.

| | Number of Students | Users | Percentage | Unique Instances of Use | Average No. of Uses |
|---|---|---|---|---|---|
| Total (2018/2019) | 644 | 113 | 17.5% | 509 | 4.5 |
| Total (2019/2020) | 764 | 270 | 35% | 1159 | 4.3 |

The third pattern of engagement centered around when the service was used. Students were more likely to use the service early in the academic year (November), when they had found their feet, and prior to submission of assignments. Although usage fell off in holiday periods (December, June, August), the service was still used. While Dollinger et al. [16] found that online students interacted with the service outside typical office hours, our study can add that students also interact with the service outside traditional study months. Many engaged with the service during traditional holiday periods when distance/part-time students often work on assignments. Support at this time was not possible through the traditional campus-based service:

> *I really appreciate you reviewing this—especially as it's over the holiday period. Thanks so much. (December 2019 WF FY)*

A spike of use in July 2020, as shown in Figure 1 over the European summer break likely relates to students repeating modules due to the impact of COVID-19.

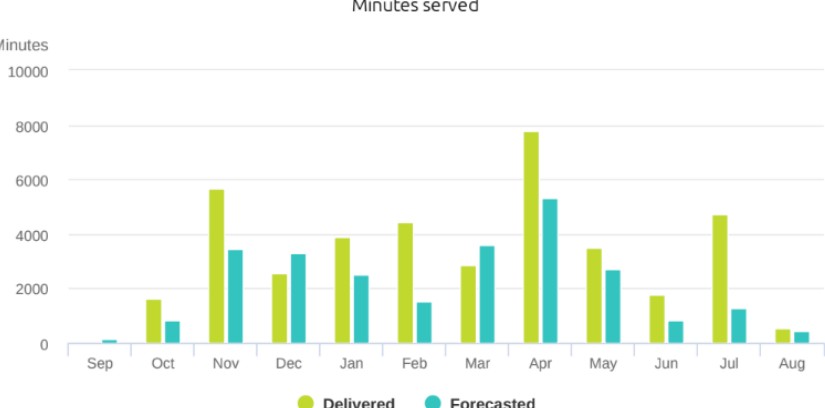

**Figure 1.** Monthly engagement 2019/2020. Note: Forecasted information was based on consultations with academic staff and the pattern of previous years' engagement.

### 4.1.2. Why Did Students Engage/Not Engage with Studiosity?

Table 4 provides a sample analysis of student question types. Regarding Writing Feedback (WF), most students appeared to be seeking help, in the form of a critical friend, to cast their eye over their work and, for the most part, determine whether they had interpreted the question correctly and answered it. Thereafter, students are mainly concerned with academic conventions, such as structure, clarity, flow, and referencing. The large number of referencing queries begs the question of whether the resources provided to students on how to reference correctly were adequate or overly complicated. While the University Library does a tremendous job of explaining why referencing is important, the numerous referencing styles seem to leave students baffled. The challenge is to make referencing more accessible for students.

**Table 4.** Sample analysis of student question types.

| Student Questions | 2018/2019 | | 2019/2020 | | Total |
|---|---|---|---|---|---|
| **Connect Live** | **PG** | **UG** | **PG** | **UG** | |
| Where is the feedback | 1 | | | | 1 |
| How to search for articles | 1 | | | | 1 |
| How to design a survey | 1 | | | | 1 |
| How to approach a question | 1 | | 5 | 8 | 14 |
| How to critique | | | | 1 | 1 |
| **Writing Feedback** | | | | | |
| Submit assignment questions for review, e.g., Have I interpreted it correctly and answered it correctly? | 25 | 27 | 17 | 26 | 95 |
| Referencing | 18 | 6 | 4 | 7 | 35 |
| Academic conventions (Structure, layout, analysis, clarity, academic language, flow, style) | 17 | 6 | 3 | 13 | 39 |

**Table 4.** *Cont.*

| Student Questions | 2018/2019 | | 2019/2020 | | Total |
|---|---|---|---|---|---|
| **Connect Live** | **PG** | **UG** | **PG** | **UG** | |
| Grammar, spelling, punctuation | 5 | 1 | 1 | 7 | 14 |
| Any help | 4 | 1 | 4 | 2 | 11 |
| Word count | 2 | 2 | | | 4 |
| English as an Additional Language learner | 2 | 8 | 2 | 4 | 16 |
| Better mark/improvement | 4 | | 1 | | 6 |
| Plagiarism | 1 | | | | 1 |
| Job reference | | 1 | | | 1 |
| Transcription service | | | 1 | | 1 |

There is evidence, in particular from first-year students' comments, that the help they received has improved their learning, showing them how to solve problems, and will be useful to them going forward:

*I found your advice on the conclusion structure a great help for this, and future assignments. Many thanks (February 2020 WF FY)*

*Always very informative. Very helpful and always seem to learn something new every time I submit a file. February 2020 WF FY)*

Connect Live (CL) interactions indicate that students were seeking a dialogue with the tutor:

*Hi This is what I have so far Stepsize = Vref/2n = 3.3/2power of 12 = 0.000806 Analogue input/stepsize = 1.65/0.000806 Ans 2047.146 This can not be correct? (May 2019 CL FY)*

There is also evidence to suggest that Studiosity attempts to engage students in dialogue in the CL sessions. Sometimes students are not interested in this approach, as the following comment illustrates:

*if a question is asked maybe rather than waiting for the student to answer it (which is impossible as it is themselves who have posed the question, why would you pose a question if you know the answer) maybe answer it. (July 2020 CL SY)*

Overall, the sense from WF and CL sessions is that the student is being provided with help, rather than answers, and that the help is thought-provoking for the student in a manner that leads to learning. The question remains whether this learning translates to tangible improvements in student achievement, a question beyond the remit of this study.

On close analysis, the questions students ask reveal that postgraduate students more regularly seek to improve their work to gain a better mark. They are less concerned about passing per se and more concerned about doing well:

*My report organisation and report formatting are less then 60%, ......what can i do to take it to the next level. (December 2019 PG WF)*

*I am trying to maintain my grade above 70% so any feedback you can offer would be greatly appreciated. (November 2019 WF PG)*

While there is nothing wrong with wanting to improve, by setting these concerns against those who are struggling to begin their higher education journey, it is difficult not to see first-year undergraduate concerns as more deserving of academic support. When first-year students are asked by Studiosity what they require help with, their answers are more concerning:

*I'm unsure at present as it is my first assignment, apologies (November 2019 WF FY)*

*I am lost in this question and just need to know how to start it (February 2020 CL FY)*

First-year undergraduate students appeared concerned with knowing what to do. While this makes sense for first-year undergraduate students, it makes little sense for postgraduate students, as illustrated in the following questions:

*Hi lads, can you check spelling, grammar also choice of language please, cheers (November 2019 WF PG)*

*I have been told by my supervisor that the questionnaire is vague. (April 2019 PG CL)*

*Hi. I am struggling with writing an academic report.... I would like to get help on how to structure my report and access resources. Thanks*

Reading between the lines, the questions from first-year undergraduate students demonstrated instrumental help-seeking, while postgraduate questions were more expedient in nature; questions which they should have been able to answer themselves, given that they have already successfully completed a programme of study in higher education.

There is also evidence that first-year undergraduate students were using the service as they lacked access to academic capital outside the institution; this type of help is often provided to more middle-class students by family members who themselves have been to university. This observation is supported by the following remarks:

*Soooooo much better than having my wife or a friend review my paper (November 2018 FY WF)*

*It's been fifteen years since I last wrote an academic essay, this advice is invaluable to me as a learner returning to education (November 2019 WF FY).*

*I am very unsure about whether this is any good at all! (November 2019 WF FY)*

*Hi, I have been told I need to provide more critique. Does this refer to:—Finding an alternative academic view on a topic or—Critique from myself, (May 2020 CL FY)*

### 4.1.3. Evaluation of Studiosity

The literature [6,16] reports high student satisfaction with Studiosity and our data is no different; students were generally delighted with the service and feedback provided to Studiosity was extremely positive:

*Wow. This is brilliant 👏👏👏 I am actually blown away by how intuitive and quick this whole process is for users. Thank you! (October 2019 WF PG)*

Students were almost always positive about the Writing Feedback (WF) service, with 79% of users stating they were Extremely Satisfied with the service in 2019/2020. Users were less positive about the Connect Live (CL) service, with just 44% stating they were Extremely Satisfied in 2019/2020. Criticism relating to the WF service related primarily to the inconsistency of feedback between reviewers, echoing the findings of Wilson et al. [7] but also indicating that students submit the same work to different reviewers:

*My issue is with inconsistencies among Studiosity feedback. (May 2020 WF PG)*

Criticism in relation to the CL service is primarily related to technical issues:

*Session terminated due to network issue at specialist end (May 2020 CL FY)*

### 4.2. Triangulating Usage Data with the Survey

The data collected by Studiosity were supplemented by a student survey provided to students in 2019. All questions are provided in Table 5, together with a descriptive analysis of quantitative data. Where relevant, responses are explained and analysed in the narrative below. Please note that not all respondents answered all questions.

**Table 5.** Survey questions and quantitative responses.

| Questions and Responses | | | | | | | Total |
|---|---|---|---|---|---|---|---|
| Q1: About the survey | | | | | | | |
| Q2: Plain Language Statement.Click to consent. | | | | | | | |
| Q3: Have you used Studiosity? | Yes<br>26 (63.41%) | No<br>15 (36.59%) | | | | | 41 |
| Q4: If you haven't used Studiosity, please indicate why not | Don't see the relevance.<br><br>1 (5.88%) | Lack of time<br><br>6 (35.29%) | Concerned about the feedback that will be received.<br>0 | Other<br><br>10 (58.82%) | | | 17 |
| Q5: How did you hear about Studiosity? | Programme Chair<br><br>16 (29.63%) | Tutor<br><br>9 (16.67%) | Fellow Student<br><br>3 (5.56%) | Moodle<br><br>21 (38.89%) | Don't know about it.<br>3 (5.56%) | Other<br><br>2 (3.70%) | 54 |
| Q6: Have you used the WF service? | Yes<br>24 (58.54%) | No<br>17 (41.46%) | | | | | 41 |
| Q7: What motivated you to use the WF service? | | | | | | | 22 |
| Q8: To what extent did the feedback you received help you to improve your assignment? | Major improvements<br><br>9 (31.03%) | Minor improvements<br><br>16 (55.17%) | Not at all<br><br>3 (10.34%) | Made worse.<br><br>1 (3.45%) | | | 29 |
| Q9: To what extent did the feedback you received help you to develop your academic writing skills? | Significantly<br><br>12 (44.44%) | To some extent<br><br>10 (37.04%) | Not at all<br><br>5 (18.52%) | | | | 27 |
| Q10: Please indicate which type of document(s) you requested feedback on. | Essay<br><br>20 (41.65%) | Case Study<br><br>6 (12.50%) | Scientific Report<br>6 (12.50%) | Thesis<br><br>5 (10.42%) | Text Analysis<br>5 (10.42%) | Other<br><br>6 (12.5%) | 48 |
| Q11: How satisfied were you with Studiosity Assignment Feedback? | Extremely Satisfied<br>16 (57.14%) | Somewhat Satisfied<br>8 (28.57%) | Neither Satisfied or Dissatisfied<br>3 (10.71%) | Somewhat Dissatisfied<br>0 | Extremely Dissatisfied<br>1 (3.57%) | | 28 |
| Q12: Have you used the CL service? If not, why? | Yes<br><br>6 (15%) | No<br><br>34 (85%) | | | | | 40 |
| Q13: How satisfied were you with the CL service? | Extremely Satisfied<br>4 (44.44%) | Somewhat Satisfied<br>0 | Neither Satisfied or Dissatisfied<br>3 (33.33%) | Somewhat Dissatisfied<br>0 | Extremely Dissatisfied<br>2 (22.22%) | | 9 |
| Q14: If you experienced any difficulties with using Studiosity please indicate what these were | | | | | | | 7 |
| Q15: Do you have any other comments to share about the value of the Studiosity service? | | | | | | | 16 |
| Q16: Any other comments? | | | | | | | 12 |
| Q17: Sex | Male<br>16 (40%) | Female<br>24 (60%) | | | | | 40 |
| Q18: Modules * | 20<br>Psychology | 10<br>Sociology | 7<br>History | 3<br>Philosophy | 2<br>Literature | 30<br>PG | 35 |
| Q19: Year of Study | 13 × 1st Y<br>(32.5%) | 5 × 2nd Y<br>(12.5%) | 1 × 3rd Y<br>(2.5%) | 4 × 4th Y<br>(10%) | 1 × 5th Y<br>(2.5%) | 15 × PG<br>(38%) | 1 × Staff<br>(2.5%) | 40 |

* Students may study more than one module

*Q4: If you have not used Studiosity, please indicate why not*

The most common (50% *n* = 9) reason given for not using the service was that they did not know about it:

*I have never heard of it before*

*Not really sure what's it's about or how to use it, not promoted or explained well enough*

While this is a small number, it seems reasonable to assume that other students were also unaware of the service. This low uptake resulted in a further, full-scale promotion of the service at this point (December 2019) to endeavour to ensure no student was left unaware of the availability of Studiosity and how it might help them.

Another interesting response was given to this question, illustrating a degree of unease about the growth of commercial third-party services:

*Encourages ill-informed interference with teaching process by commercial agents unscreened by local educational professionals*

This comment came from a tutor respondent. Studiosity was designed to supplement existing campus-based services rather than replace them. However, a small number of tutors expressed concern at the introduction of the service and were worried that it would interfere with the teaching and learning relationship they had with students, an experience also outlined in the literature [6]. Despite assurances that the service would not provide students with answers, a few tutors remained somewhat hostile and wary of Studiosity. This might partly explain why some students did not use the service. DCU's Open Education modules are supported through a wide network of tutors operating on one-year contracts which are renewed annually. The precarious nature of their employment may have sharpened their hostility to introducing the third-party academic support service. An investigation of this theme is beyond the scope of this research but is an area worthy of further study.

Q5: How did you hear about Studiosity?

In relation to the above question, the 'Other' response was as follows:

*An aggressive campaign of contact from the Studiosity team*

Studiosity enthusiastically promoted its service to users with permission from the institution. Tutors, too, were asked by the institution to promote the service to students. Time was spent in assuaging tutor concerns around the service. However, the importance of orientating tutoring staff and those who support teaching to the service cannot be overstated [6].

Q7: What motivated you to use Studiosity?

While responses to this open question very much tie in with the analysis of student questions in the Studiosity Dataset (i.e., references, grammar, structure, etc.), the survey added another layer to our understanding of why students used the service, relating to an understandable desire among new students to receive formative feedback before submission for summative feedback.

Distance undergraduate students new to university study often have few resources to draw on when it comes to having someone check their work before submission. While distance education tutors are tasked with providing comprehensive feedback to students, this is normally completed in a summative manner, though sometimes tasks are so lightly weighted they are, in effect, formative. The following comment indicates the desire for formative feedback:

*Never having written academic essays before, I really appreciated any assistance that would improve my work. Although our tutors give a broad outline on what is required, they are unable to give feedback on draft essays.*

The responses from post-graduate (PG) students had a slightly different emphasis. As with the Studiosity data, the focus was to enhance their grades:

*I want to use all tools available to help me improve my grades.*

In this respect, these students were making strategic use of the service. For some students, the fact that the person reviewing their work would not be grading it was important:

*I was curious and in need of an unbiased opinion.*

*It's good to get peer review from someone who does not know you and will be impartial*

Q12: Have you used the Studiosity online chat service (Connect Live)? If not, why not? (*n* = 19)

The most common reason (*n* = 9) for not using the service was that respondents felt they did not need it, with two respondents (*n* = 2) stating a preference for the written feedback service, one respondent (*n* = 1) stating they did not have the time to use the Connect Live service and one (*n* = 1) stating that their broadband connection was not strong enough. Six respondents were unaware of the service. The real-time nature of the CL service, coupled with the fact that Ireland has a somewhat patchy broadband service outside major cities and towns, seems to have directed students away from the CL function.

Q14: If you experienced any difficulties with using Studiosity, please indicate what these were.

Most respondents (*n* = 5) did not have any difficulties using Studiosity. The other two responses, one from a tutor, expressed concern about the lack of expert knowledge:

*The 'expert' had no knowledge of any aspects I asked about. No knowledge about crtical theories and wasted the allocated time asking me stupid questions.*

*Studiosity adopts a one-fits-all approach to essay writing that is not discipline specific.*

This response echoes to some extent research from Benzie and Harper [10] (p. 645) who argue that third-party support services 'can complicate the development of student writing in higher education by asking learners to engage in a process of trial and error in order to piece together writing advice from different, external sources that may offer inaccurate, irrelevant or conflicting information'.

Q15: Do you have any other comments to share about the value of the Studiosity service?

As with the Studiosity data, respondents (*n* = 15) were mostly positive (*n* = 11) about the service:

*Studiosity service is great. It is quick and reliable when it comes to feedback. It gives feedback within 24 hrs. It ensures that the work is academic standard. give honest comment and giving option to go back to correct them before submitting your work.*

Despite this positive feedback, one comment indicates that tutors may have passed on their concerns about Studiosity to students:

*There was a misconception with tutors that there would be subject advise but Studiosity gave me more confidence that what I was trying to say was being said, rather then being lost in bad writing. It was like a human grammarly and didn't comment on the subject matter at all. Absolutely fantastic service that I would highly recommend to everyone.*

Some respondents (*n* = 3) were more guarded in their praise:

*As with all services, the quality of feedback varied depending on who was assessing my work. Some 'tutors' were very thorough offering invaluable feedback and suggestions and others were not so good. However on one occasion I submitted the same essay twice which really benefited me.*

Finally, one comment from the tutor respondent was as follows:

*It is irrelevant, unhelpful and unwelcome, possibly even disruptive of student-teacher relations*

Q16: Any other comments?

This question was designed to unearth any questions participants might have which related to Studiosity but not the service per se. Again, most responses were positive (*n* = 9). One comment indicated the importance of this service to off-campus students:

*I hope [institution] continue to offer students this service as it has not only helped me to improve my work but it has given me more confidence in my own ability. This service is invaluable to "connected"/online students as interaction with our tutors and other students is quite minimal. Also although [institution] does offer the facilities of its writing centre, it's hours are limited and don't always suit online students*

However, two comments (*n* = 2) were negative, with one seeming to support tutor concerns regarding interference in the teaching process:

*I am baffled how we are encouraged by the powers that be in [institution] to avail of the service but the tutor doesn't want it used—for the obvious reason that some of the feedback was total rubbish i.e., in an English lit dissertation feedback I was told to write as if the reader had no knowledge of the novel which is in direct contrast to feedback from tutor.*

And the following was the final comment (from a tutor):

*End the contract*

This final comment, echoing the findings of Wilson et al. [6], raises the question of whether tutors are uncomfortable with the development of online student support and development services—regardless of whether they are provided in-house or through a third-party provider. There are wider implications for the effective implementation of these services, irrespective of the provider. This too is worthy of further research.

## 5. Synthesis of Key Findings

This study set out to examine online academic help-seeking through outsourced support in a university context. The aim was to better understand the pattern of student cohort engagement with Studiosity and the implications of this pattern. Ancillary questions related to why students engaged/did not engage with Studiosity and their perceptions of its use. This section provides a brief synthesis of the key findings.

*What was the pattern of student cohort engagement with Studiosity and what are the implications of this pattern?*

We found that postgraduate students were strong users of the service, using it more than first-year undergraduates. This is an important contribution as it indicates that, all other things being equal, those with high levels of academic capital, who arguably need an academic help service least, may use it most. They were highly strategic in their use of Studiosity, with the goal of increasing their marks and grade-point average.

But first-year undergraduate online distance students were also strong users of the service. This is also an important contribution. The literature suggests that undergraduate students, in particular those from underrepresented groups, favour less formal help-seeking [3,17,32]. Studiosity was not informal; it was provided through the institution. What seemed to be most important to students was that the academic support was well publicised, readily available to them, easy to use, and independent of their academic programme and of their tutor/professor.

*Why did students engage/not engage with Studiosity (what can we learn from the questions they ask)?*

The help-seeking behaviour of the postgraduate cohort tended to be expedient in nature, as they sought answers to questions they should, given their postgraduate status, have been able to answer, or source the answer to, themselves. This finding is also an important contribution to the literature. It raises the question of the value of providing expensive academic support to a cohort who may not require it and who traditionally do not have high rates of attrition.

The help-seeking of the first-year undergraduate cohort was instrumental in nature, as they sought to problem-solve and improve their learning in a way that would help them move forward and persist. Some questions were concerned with simply knowing what to do, but this makes sense for first-year undergraduates in a way that it does not for postgraduates. Finally, their questions often indicated a dearth of background academic

capital. This is a problem that universities must endeavour to address. Undergraduate students who successfully complete the first year are more likely to persist [45], so targeting first-year students with academic support makes sense.

The reason students gave for not using Studiosity was that they did not know about it. New students are often overwhelmed with information and are not always aware of important support services. Proactive promotion of academic support, at key times and to those who need it most, seems critical to success. This might mean giving appointments to this cohort in the first instance, in order to develop their help-seeking practice, rather than waiting for them to make an appointment.

There was little evidence in this study that students were simply provided with answers, or that the service was primarily an editing one.

*What were students' perceptions of Studiosity?*

Students were overwhelmingly positive about Studiosity. They saw it as readily available, easy to access and the help as objective and valuable to their learning and development. The data in this paper demonstrate the importance of online academic support services for students who may be online, part-time, distance or working, in particular, at key points in the academic year.

These contributions are important as they add to our understanding of academic help-seeking behaviour and can help inform academic support provision at university level.

## 6. Conclusions

Prior to the pandemic, the University's Writing Centre provided an average of 180 min per week of online support for all university students. However, the demand for Studiosity services during that period, from distance students alone, was around 400 min per week (2018/2019). This demand rose to an average of 975 min per week during the pandemic (2019/2020). The COVID-19 crisis saw the university pivot to a 'digital-first' provision, leading to the development of online learning support, which has persisted. For this, among other reasons, including the challenge of meeting procurement requirements for public institutions beyond a small pilot, the university no longer deploys Studiosity.

In the final analysis, academic help is important to student retention and success. Whether or not it is worthwhile for a university to use a third-party provider like Studiosity is a difficult question to answer. The nature of the relationships between higher education institutions and commercial companies to whom they seek to outsource key support functions, particularly of a teaching nature, are complex and can be contentious. There are costs, trade-offs, and tensions inherent in these relationships.

The wider context of online learning is increasingly complex. New generative Artificial Intelligence (AI) platforms are changing the tools available and the wider student help-seeking landscape. Effective academic support and development across the study lifecycle is costly to provide. Service providers such as Studiosity work on the basis that they can provide a service to multiple users in a scalable and more cost-effective way than would be possible otherwise. Reading between the lines, the evidence from our research indicates that not all students need academic support equally. First-year undergraduate students appear to use academic help in an instrumental manner, in an effort to persist with their studies. Institutions would be wise to develop their learning support ecosystem in a manner that targets students at risk of attrition and who are most likely to benefit over their entire learning journey.

## 7. Limitations

There are limitations with this study. The findings relate to just one institution, so are illustrative rather than exhaustive. The data were collected at a particular point in time, with a particular cohort of students, and will not generalise to all times and all students. The coding of content segments requires interpretation. To this extent, it is important to point out that we are presenting our interpretation of the qualitative data, albeit informed by robust methodological procedures.



### 8. Further Research

There are several areas relating to this topic which are worthy of further research. Firstly, it would be useful to explore staff perceptions of academic support, whether this differs between those who hold permanent jobs against those in more precarious employment, and whether it differs according to university or third-party support provision. Secondly, given the datasets available in universities, it would be useful to explore a causal relationship between academic support and retention. Finally, it would be interesting to explore student use of Generative AI for academic support.

**Author Contributions:** Conceptualization, L.D.; methodology, L.D.; software, M.B. and E.C.; validation, L.D., M.B. and E.C.; formal analysis, L.D.; investigation, L.D.; resources, L.D.; data curation, L.D., M.B. and E.C.; writing—original draft preparation, L.D.; writing—review and editing, L.D., M.B. and E.C.; visualization, L.D.; supervision, L.D. All authors have read and agreed to the published version of the manuscript.

**Funding:** This research received no external funding.

**Institutional Review Board Statement:** Ethical approval was obtained from the university ethics committee (DCUREC/2019/075).

**Informed Consent Statement:** Informed consent was obtained from all subjects involved in the study.

**Data Availability Statement:** Data is unavailable due to privacy or ethical restrictions.

**Conflicts of Interest:** The authors declare no conflict of interest.

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
