# Peer review of "Between the Lines: An Exploration of Online Academic Help-Seeking and Outsourced Support in Higher Education: Who Seeks Help and Why?"

_education, doi:10.3390/educsci13111147_

Round 1
Reviewer 1 Report
Comments and Suggestions for Authors
Dear Authors
I converted the PDF copy of your paper into a Word document, made comments in that, and converted it back to a PDF file.
The topic is current and of interest to universities that provide academic support services to students. The research conducted at your university adds value to understanding how Studiosity was utilised in your context.
The literature review, methodology, discussion of findings, and implications for practice all need additional work. The article requires more analysis in some areas. I think the way you present your information can be improved.
I have suggested changes across several of these areas using the Comments function.

Author Response
We would like to thank the reviewer most sincerely for the time taken to provide such comprehensive and valuable feedback on our paper. We have endeavored to address all feedback in detail. Our response to each comment is outlined below.
Response to Reviewer 1.
Line 21: Change accepted
A1 line 29: We have removed this comment/reference.
A2 line 33: Change accepted
A3 line 41: We have removed this comment/reference.
Line 52: Change accepted
A4-5 Line 61-73: We have put this information into a table and included some explanatory narrative which hopefully makes the section clearer.
A6 line 90-97: We have revised this paragraph to pack more of a punch (we hope).
A7 The Literature Review: We have restructured this chapter in line with the feedback.
- Provided a brief introductory paragraph outlining what we are going to cover in the review.
- Re structured the chapter so that it has a more logical flow.
- More clearly identified key points of critique as follows:
- We question the taken-for-granted legitimacy of academic support provision to help postgraduate students transition to their studies.
- Related to the above point we highlight that there is no evidence of high attrition among post-graduate students.
- We highlight the limitations of peer support, which is often highlighted as appropriate support for underrepresented students in the literature.
- We have called out the potential importance of expedient help seeking for those who do not enter university with background academic capital.
- We identify similarities and differences between existing research in relation to:
- Studies which focus on academic support to enhance persistence of underrepresented undergraduate students against literature that focuses on academic support for all students equally.
- Studies which focus on campus based in person support against those that focus on online support and/or third-party support.
- Studies that focus on instrumental versus expedient help-seeking
- More clearly identified the gaps in the literature in the concluding paragraph as follows:
- Whether help seeking through Studiosity is instrumental or expedient
- Research on the extent to which help-seeking differs between first year undergraduate and postgraduate students.
A8 line 203: We have changed this bullet point to specify student cohorts. We have identified the final bullet point as a gap in the literature review section.
A9 line 270: This section is now titled Pattern of student cohort engagement with Studiosity and discusses various patterns, identifying differences between first year undergraduate and postgraduate students.
A10 line 300-302. We confirm this finding was also identified by Dollinger et al.
A11 line 318: We have identified the literature which reports high student satisfaction ratings with Studiosity.
A12 Survey: We have inserted a Table in the paper outlining the survey questions and quantitative responses. The narrative comments on the issues arising.
A13: The survey questions are now provided in the table.
A14: We have, we hope, more clearly outlined how the data answers the research questions and what stands out.
A15: We have, we hope, more clearly analysed our results in relation to issues raised in the literature. While we think it is important for universities to decide for themselves whether or not it is worthwhile to use a provider like Studiosity, we have made our experience clear to the extent that we found the service costly and academic support can be effectively provided in a more tailored manner. We have, we hope, made our contribution more clear.
Reviewer 2 Report
Comments and Suggestions for Authors
Student support through the provision of educational platforms is a topical issue, and its importance could be appreciated during the COVID-19 pandemic.
The results of the study presented by the authors add to the knowledge about the use of one of the platforms supporting education in synchronous and asynchronous modes - Studiosity.
The article lacks some elements by which it should be supplemented to meet the standards of scientific papers.
I have the following specific comments and suggestions:
1. I believe that most of the content from the introduction, especially describing the student groups studied and the Studiosity platform, should be moved to the empirical part, i.e. after point 2 "Literature review". Instead, the introduction should end with a brief description of the scope of the article.
2. Sentence (?) "argue that an effective online learning ecosystem depends on this type of planning and support for distance learners. (page 1 rows 41-42)" is incomprehensible. What "type of planning?")
3. Abbreviation "OEU" (page 2 row 53) should be clarified.
4. There is a lack of discussion of the results of the survey conducted against the background of previous studies.
5. Either in the conclusion or in separate paragraphs, the limitations of the study and directions for further research should be indicated.
6. The data analyzed in the article is from 2019-2020. do the authors not have more recent data? If not, why did they prepare a peer-reviewed article only now in 2023 ?
7. More recent data - from the period of the pandemic - would certainly increase the value of the reviewed work.
8. Please consider whether the results described in section 4.2 would not be better presented in a table ?
The above comments indicate the need for some corrections and additions to the reviewed article.
Author Response
We would like to thank the reviewer most sincerely for the time taken to provide such comprehensive and valuable feedback on our paper. We have endeavored to address all feedback in detail. Our response to each comment is outlined below.
Responses to comments from Reviewer 2
- The sections describing student groups and the Studiosity platform have been moved to the Methodology section. The introduction concludes with a brief description of the scope of the article.
2. This sentence has been removed.
3. OEU has been clarified (Open Education Unit).
4. We have expanded the discussion of survey results, particularly for questions 14 and 16, providing a background of previous studies.
5. We have expanded on the limitations of the study at the end of the Methods section and outlined directions for further research after the Conclusion.
6. The university did not continue to use Studiosity beyond the identified time frame. There were many, both personal (related to the Pandemic) and work-related reasons, why it has taken until now to present our findings for publication. However, we entirely accept this point and have identified the time-frame as a limitation of the work at the end of the Methods section.
7. We hope that the main findings of our research are not time dependent and rather help address a gap in the literature on our understanding of academic help-seeking. We think this may be particularly relevant as universities adapt their student support services for the challenges and opportunities of GenAI.
8. We have presented the survey results in a table.
Round 2
Reviewer 2 Report
Comments and Suggestions for Authors
Thank you for considering my comments and suggestions.
However, I believe that according to the generally applied rules for the development of scientific articles, the limitations of the study should be presented at the end of the article and not in the methods section.
Having taken this comment into account, I believe that the article in its revised form deserves to be published in the journal Education Sciences.
Author Response
Response to Review 2 (Round 2)
We would again like to thank the reviewer most sincerely for the time taken to attend to the detail of our paper. It is very much appreciated.
Response
We have removed the limitations of the study from the Materials and Methods section and placed it at the end of the article under the heading Limitations.